# Photochemical Generation of Methyl Chloride from Humic Aicd: Impacts of Precursor Concentration, Solution pH, Solution Salinity and Ferric Ion

**DOI:** 10.3390/ijerph17020503

**Published:** 2020-01-13

**Authors:** Hui Liu, Yingying Pu, Tong Tong, Xiaomei Zhu, Bing Sun, Xiaoxing Zhang

**Affiliations:** College of Environmental Science and Engineering, Dalian Maritime University, Dalian 116026, China; py1120181596@dlmu.edu.cn (Y.P.); tongtong@dlmu.edu.cn (T.T.); zhuxm@dlmu.edu.cn (X.Z.); sunb88@dlmu.edu.cn (B.S.); zhangxiaoxing@dlmu.edu.cn (X.Z.)

**Keywords:** methyl chloride, humic acid, photochemical production, ferric ion

## Abstract

Methyl chloride (CH_3_Cl) is presently understood to arise from biotic and abiotic processes in marine systems. However, the production of CH_3_Cl via photochemical processes has not been well studied. Here, we reported the production of CH_3_Cl from humic acid (HA) in sunlit saline water and the effects of the concentration of HA, chloride ions, ferric ions and pH were investigated. HA in aqueous chloride solutions or natural seawater were irradiated under an artificial light, and the amounts of CH_3_Cl were determined using a purge-and-trap and gas chromatography-mass spectrometry. CH_3_Cl was generated upon irradiation and its amount increased with increasing irradiation time and the light intensity. The formation of CH_3_Cl increased with an increase of HA concentration ranging from 2 mg L^−1^ to 20 mg L^−1^ and chloride ion concentration ranging from 0.02 mol L^−1^ to 0.5 mol L^−1^. The photochemical production of CH_3_Cl was pH-dependent, with the highest amount of CH_3_Cl generating near neutral conditions. Additionally, the generation of CH_3_Cl was inhibited by ferric ions. Finally, natural coastal seawater was irradiated under artificial light and the concentration of CH_3_Cl rose significantly. Our results suggest that the photochemical process of HA may be a source of CH_3_Cl in the marine environment.

## 1. Introduction

Methyl chloride, CH_3_Cl, is particularly important in the global atmosphere as a major natural source of chlorine and plays a significant role in atmospheric chemical processes such as stratospheric ozone depletion [1]. Besides, as an absorber of infrared radiation, CH_3_Cl is of interest for its potential effect on the tropospheric energy balance [2]. The Global Warming Potential (GWP) of CH_3_Cl is estimated to be about eight (100-year time integration, CO_2_ = 1), indicating that CH_3_Cl has a direct GWP similar to that of CH_4_ [3]. The production of CH_3_Cl is dominated by natural sources, but smaller, important anthropogenic sources, such as agricultural fumigation and/or biomass burning, also exist [4]. Natural emissions of CH_3_Cl include oceanic sources, terrestrial plants, fungi and biomass burning, etc. [5]. Among these, the ocean has been implicated as one of the main natural sources, where inorganic halogen ions in seawater or its aerosol can transform into organic halogen through biotic or abiotic pathways. There are several reports that organisms such as macroalgae, microalgae and bacteria can release large quantities of halogen hydrocarbons into the atmosphere [6]. However, few studies have been focused on the abiotic sources of halogen hydrocarbons in the ocean.

Photochemical processes occur universally within surface seawater and are considered to be an important abiotic pathway for the formation of organic halogens [7]. There are a number of reports on the photochemical production of methyl halides (chiefly methyl iodide) in seawater [8,9]. It is supposed that a component of light absorbing molecules, such as marine colored dissolved organic material (CDOM), takes part in the formation of methyl halides. Dissolved organic matter (DOM) in estuaries and coastal areas is a class of natural organic compounds containing molecules derived from biota in terrestrial (allochthonous) and/or marine (autochthonous) ecosystems [10], which contains a large assemblage of complex chemical structures of polyphenols, carboxyls, methoxyls, quinones, carbohydrates and peptide functionalities [11]. As a complex macromolecular mixture, DOM can absorb light energy and induce photochemical reactions. DOM is known to act as the sensitizer or the precursor for various reactive species in aquatic ecosystems, such as hydroxyl radicals (·OH), singlet oxygen (^1^O_2_) and the excited triplet state of DOM (^3^DOM*), which then participate in the photochemical transformation of organic and inorganic compounds in water [10,12,13].

It is notable that DOM plays an important role in the formation of organic halogens from halide ions and non-halogen compounds. The excited state of DOM, forming upon sunlight irradiation (especially ^3^DOM*), can oxidize halide ions to generate reactive halogen species, such as X_2_^•−^/X^•^, X_2_ and HXO (X = Cl, Br or I), which are the major halogenation reagents [13,14]. In addition, DOM itself can be halogenated, forming non-volatile and volatile org-halogens, i.e., DOM acts as the precursor of organic halogens [15,16]. Research on the photochemical formation of volatile org-halogens in saline water have been concentrated on methyl halogens. Moore [9,17] and Yang et al. [18] have reported that lignin model compounds, such as syringic acid and guaiacol, could act as the carbon source to produce CH_3_Cl in saline waters through a photochemical pathway. However, little study was focused on the formation of CH_3_Cl from natural DOM. Natural DOM is a complex mixture containing humic acid-like and fulvic acid-like materials, and soluble microbial byproduct-like materials; DOM in coastal and estuarine environments is even more versatile [19]. Therefore, the generation of CH_3_Cl from its natural complex precursor in the variable water environment needs further investigation.

In estuarine and coastal waters, due to the mixing of river water (which is rich in terrestrial DOM) and seawater (which has high salinity), sharp increases in DOM and chloride ions occurs simultaneously, which offers a great potential for the production of organic halogens [20]. Considering that humic acid (HA) is a typical terrestrial DOM in water, and HA can stabilize metal ions, e.g., iron, which plays an active role in the photochemical reactions [21], HA was selected as the model DOM in this work. The primary objective of this work was to identify the formation of CH_3_Cl from HA in saline solutions under simulated sunlight irradiation, and the influence of some environmental factors on CH_3_Cl generation, including chloride ion concentration, sunlight intensity, pH and ferric ion (Fe (III)). Furthermore, the production of CH_3_Cl from natural seawater was investigated as well.

## 2. Materials and Methods

### 2.1. Reagents

HA was supplied by MP Biomedicals, Inc (Eschwege, Germany). Liquid standards of CH_3_Cl (200 μg mL^−1^ in methanol) were purchased from AccuStandard (USA). Sodium chloride (NaCl), FeCl_3_·6H_2_O and other chemicals were reagent grade. Ultrapure water (18 MΩ cm) was obtained from a Millipore water purification unit for preparing all aqueous solutions. Natural coastal seawater was collected from the Northern Yellow Sea near Blackrock Reef (38°55′ N, 121°37′ E), Dalian, China.

### 2.2. Irradiation Experiments

The reactor used in the experiment was a round bottom sealed quartz tube (3.5 cm o.d.; volume approximately 125 mL) with one outlet (4 mm o.d.) at the middle of the bottom of the tube, where the reaction solution was injected into the reactor by a syringe. The quartz tube reactors were checked for air tightness before irradiation experiments. All quartz tubes were irradiated in a solar simulator (Phchem III, Beijing Newbit Technology Co., Ltd., Beijing, China) containing a 500 W xenon arc lamp and filters to cut off the light with a wavelength below 290 nm. The light intensity of the solar simulator can be adjusted from 3 mW cm^−2^ to 15 mW cm^−2^. Details are in Appendix A. The aqueous solutions of HA (2–20 mg L^−1^) and chloride (0.02–0.5 mol L^−1^) were held in quartz tubes and irradiated for periods of 20 h. The quartz tubes were cooled by a fan during irradiation; the dark controls in tubes wrapped in Al foil were also placed in the solar simulator. All experiments were run in replicate. The light intensity effect was first studied at 3.5 mW cm^−^^2^ and 12 mW cm^−^^2^, separately. Light intensity was set at 12 mW cm^−2^ for all other experiments. Then, the concentration of HA, Cl^−^, ferric ions and pH on the production of CH_3_Cl were separately investigated. Finally, the generation of CH_3_Cl in natural seawater was measured.

### 2.3. Analysis Methods

The concentration of CH_3_Cl was analyzed using a gas chromatography mass spectrometer (GC-MS, Agilent 7890B/5977C, Agilent Technologies, Santa Clara, CA, USA) equipped with a purge-and-trap sample concentrator (Eclipse 4760, OI Analytical, College Station, TX, USA). After irradiation, aqueous samples were withdrawn from the tubes by a glass syringe via a long needle, and then injected into a 25 mL purge tube. Before purging, the tube of each sample was thermostated at 40 °C. After that, the sample was purged with ultrapure nitrogen at 40 mL min^−1^ for 11 min to facilitate the reproducible extraction of trace gases. Extracted gases were pre-concentrated in a collecting trap of a stainless-steel tube containing Vocarb-3000. CH_3_Cl was released from the trap column by heating the trap to 240 °C for 1 min, and then the gas was introduced into the GC-MS. After each capture and desorption procedure, the trap column was baked at 250 °C for 10 min to avoid interference between samples. The temperature of the transmission line between the purge-and-trap concentrator and the GC-MS was kept at 200 °C. An Agilent J&W DB-VRX capillary column (length, 60 m; inner diameter, 250 μm; film thickness, 1.4 μm; Agilent Technologies, Palo Alto, CA, USA) was used for the analysis of CH_3_Cl. The carrier gas was helium with a flow rate of 1 mL min^−1^. The inlet worked in a split mode with a split ratio of 20:1 and the temperature was set at 150 °C. The oven temperature was initially set at 32 °C for 6 min and rose to 180 °C at 20 °C min^−1^ then held at 180 °C for 4 min. An illustration of the purge-and-trap device and the GC-MS is shown in Appendix A. CH_3_Cl was quantitatively analyzed by direct external calibration. The selected ion monitor (SIM) mode was used with an *m*/*z* of 50 and 52. The detection limit (signal to noise ratio, S/N = 3) of CH_3_Cl was 3.2 ng L^−1^, and the relative standard deviation of replicate analyses (*n* = 6) of CH_3_Cl was within 6%. For qualitative analysis of other volatile gases from HA, scan mode was used with an *m*/*z* from 35 to 350, and the structure was identified by comparison with the National Institute of Standards and Technology (NIST) 2014 Library.

### 2.4. Characteristics of HA and HA-Fe (III)

The elemental composition of HA was analyzed on an element analyzer (Vario EL III, Elementar Analysensysteme GmbH, Langenselbold, Germany). FTIR spectra of HA and the HA-Fe (III) complex were recorded with an FTIR spectrometer (Nicolet iS 5, Thermo Fisher Scientific, Madison, WI, USA).

## 3. Results and Discussion

### 3.1. Influence of Light Intensity

It has been found that the formation of methyl iodide under light irradiation was more obvious than its formation in the dark [22], showing that sunlight irradiation is crucial for the production of methyl halide in seawater. The influence of light intensity on the generation of CH_3_Cl was first investigated in the solutions containing HA and Cl^−^. DOM as a part of the total organic carbon (TOC) in the open ocean is extremely low, whereas it is much higher in estuaries which can reach dozens of mgC L^−1^. The elementary composition for HA used in the work is 40.8% C, 3.0% H, 50.6% O and 0.9% N. HA at a concentration of 5 mg L^−1^ with the corresponding TOC at 2.04 mgC L^−1^ was first adopted in this experiment. The sunlight intensity of the water surface is about 85 mW cm^−2^ at noon on a sunshine day in Dalian, China. Considering the severe attenuation of light in water and the changes of the solar altitude angle, the average light intensity in the euphotic zone of the water body should be much lower. Thus, the light intensities were set at 3.5 mW cm^−2^ and 12 mW cm^−2^ in this work. It can be seen that the generation of CH_3_Cl was significantly increased with increased light intensity and was not detectable in the dark within 20 h (Figure 1). This result is similar to the finding of Moore that CH_3_Cl from syringic acid was only produced under illumination [9].

### 3.2. Influence of HA Concentration

Many reports have confirmed the role of DOM in controlling halogen hydrocarbon production in seawater. Manley and Barbero demonstrated that the removal of DOM from natural seawater decreased CHBr_3_ production from *Ulva lactuca* sp. [23]. Lin and Manley found a significant correlation between DOM molecular size and CHBr_3_ production from seawater treated with bromoperoxidase [24]. The following experiments were performed to study the effect of HA concentration on the production of CH_3_Cl through photochemical processes. The concentration of HA, ranging from 2 mg L^−1^ to 20 mg L^−1^, almost covers the DOC concentration in coastal and estuarine water. It can be seen from Figure 2A that the amount of CH_3_Cl increased with increasing HA concentration. When the HA concentrations were 2 mg L^−1^, 5 mg L^−1^, 10 mg L^−1^ and 20 mg L^−1^, the maximums of CH_3_Cl were 43.5 ng L^−1^, 49.2 ng L^−1^, 56.5 ng L^−1^ and 68.3 ng L^−1^, respectively. However, the formation of CH_3_Cl in the presence of 20 mg L^−1^ HA was not obviously higher than the lower HA concentrations within 10 h of irradiation. The reason for this may be related to the light shielding effect of HA, which can reduce the absorbance and the effective utilization of photons.

In order to estimate the light shielding effect of HA on CH_3_Cl production, the light shielding correction factor of HA (CF_HA_) is calculated according to the following equations [25,26]:(1)$${\mathrm{CF}}_{\mathrm{HA}}=\frac{\sum_{\lambda }I_{0,\lambda }a_{\lambda }}{\sum_{\lambda }I_{0,\lambda }a_{\lambda }S_{\lambda }}$$
(2)$$S_{\lambda }=\frac{1-10^{\left(-a_{\lambda }\right)z}}{2.303a_{\lambda }z}$$
where *I_0,λ_* is the initial irradiation intensity at wavelength λ as displayed in Appendix A; *S_λ_*, the light shielding factor, is determined by Equation (2) where *z*, and *a_λ_* (cm^−1^) refer to light path length and light attenuation factors of HA at wavelength λ, respectively. In this study, CF_HA_ was calculated for the wavelength range of 300–600 nm, wherein HA exhibits considerable absorbance (Appendix A). The CF_HA_ is 1.04, 1.10, 1.18 and 1.36 for 2 mg L^−1^, 5 mg L^−1^, 10 mg L^−1^ and 20 mg L^−1^ HA, respectively. The concentration of CH_3_Cl generated after corrected for light shielding, C_CH3Cl_^’^, is calculated according to Equation (3).
(3)$$C_{\mathrm{CH}3{\mathrm{Cl}}^{’}}=\;C_{\mathrm{CH}3{\mathrm{Cl}}^{}}\times {\;\mathrm{CF}}_{\mathrm{HA}}$$

The production profile of CH_3_Cl after being corrected for light shielding is shown in Figure 2B. The production rates of CH_3_Cl in the presence of different concentrations of HA at 10 h were estimated and displayed in Figure 3. It can be seen that the rate constant (k) of CH_3_Cl formation after it was corrected for light shielding, k_CH3__Cl_, was almost linear with HA concentration, which indicated that HA acted as the precursor of CH_3_Cl and was the limiting factor under this experimental condition.

However, CH_3_Cl production was slow after long term irradiation (>10 h). The reason for this is most likely related to the slow formation of the active precursor of CH_3_Cl after long term irradiation. Moore has demonstrated that the photochemical production of CH_3_Cl from syringic acid was not directly from the acid itself, but a product of its photolysis (most likely a quinone derivative) [9]. Unfortunately, the nature of this intermediate is unknown. The structure and composition of HA are more complex than that of syringic acid, and thus the situation in this experiment will be more complicated. We proposed that the postulated intermediate that was liable for CH_3_Cl production might be generated fast in the first period of irradiation, and then its generation became slow after long term irradiation. Additional work is needed to elucidate the nature of the active intermediate of CH_3_Cl production from HA.

### 3.3. Influence of Chloride

As a variable element in the estuarine area, the concentration of chloride ions in water often affects the results of photochemical reactions [27]. The salinity of sea water around the world is variable with an average value of 34.7, and the salinity range of estuaries is extremely large depending on the mixing of seawater and river water, which is influenced by many factors such as evaporation, precipitation, river runoff and seawater movement. These experiments were carried out with the concentration of Cl^−^ ranging from 0.02 mol L^−1^ to 0.5 mol L^−1^ with the corresponding salinity of 1.3‰ to 32.4‰, which can be applied to mimic the salinity changes in estuarine and coastal regions. When Cl^−^ concentrations were 0.02 mol L^−1^, 0.2 mol L^−1^ and 0.5 mol L^−1^, the concentrations of CH_3_Cl after 20 h of irradiation were 24.7 ng L^−1^, 29.5 ng L^−1^ and 49.2 ng L^−1^, respectively (Figure 4), indicating that Cl^−^ was the other precursor of CH_3_Cl. Estuarine and coastal waters are rich in terrestrial HA and chloride ions; thus, it can be speculated that the photochemical generation of CH_3_Cl could occur in estuaries and coastal areas. However, promoting CH_3_Cl production by increasing chloride was not as obvious as increasing HA concentration, especially in the first 5 h. We proposed that the limiting factor for CH_3_Cl production in our reaction systems is HA, not chloride. Thus, there was no significant increase in CH_3_Cl in the first 5 h of irradiation in the presence of different concentrations of Cl^−^.

### 3.4. Influence of pH

HA comprises multiple aromatic and aliphatic hydrocarbon structures, and its configuration is often distinct at different pH levels [28,29]. The pH of the solutions containing 5 mg L^−1^ HA and 0.5 mol L^−1^ Cl^−^ was adjusted using hydrochloric acid or sodium hydroxide from 4.0 ± 0.1 to 10.0 ± 0.1. Figure 5 shows that the concentration of CH_3_Cl at pH 4.0 (26.6 ng L^−1^) was significantly lower than at pH 6.5 (around 45 ng L^−1^). As pH increased from 6.5 to 10.0, CH_3_Cl formation decreased. The order of CH_3_Cl concentration was pH 6.5 > pH 8.0 > pH 10 > pH 4.0, indicating that the most suitable conditions for CH_3_Cl generation from HA were near neutral conditions. According to Moore [9,17] and Yang et al. [18], the methoxy group in simple lignin-like molecules can be the source of the methyl group in CH_3_Cl produced by a photochemical reaction. HA is a complex mixture partially deriving from the lignin structures of higher terrestrial plants [11] and consequently contains syringyl and guaiacyl lignin groups. The FTIR spectrum of HA (Figure 6) shows the following characteristics: -OH stretching vibration absorption (ν_O–H_) at 3700–2700 cm^−1^, C-H stretching vibration absorption of aliphatic series (ν_C–H_) at 2923/2851 cm^−1^, carbonyl C=O absorption (ν_C=O_) around 1563 cm^−1^, bending vibration of alcohols or carboxylic acids (δ_O–H_) and stretching vibration of phenols (ν_C–O_) around 1369 cm^−1^; C-O stretching vibration of alcohols, ethers, phenol, and/or polysaccharides around 1100–1200 cm^−1^ [30,31]. Thus, it can be seen that HA contains functional groups such as carboxyl, hydroxyl, methoxy and phenolic groups.

According the previous studies on CH_3_Cl from lignin-like compounds, the proposed formation pathway was via aromatic ring protonation followed by demethoxylation [9,17,18]. We took the syringyl moiety in HA as an example. The first step was the excitation of the syringyl moiety by the absorption of photons and its subsequent protonation on the benzene ring. The second step involved a nucleophilic attack of Cl^−^ on the methoxy carbon of the syringyl moiety resulting in C_methyl_–O cleavage and CH_3_Cl releasing (Figure 7).

In the pH range of 6.5 to 10, the fraction of syringyl moieties presenting in their anionic form increased with increasing pH, given that the pKa2 of syringic acid is 9.45 for the ionization of phenolic hydroxy (pKa1 of syringic acid is 4.33 for the ionization of carboxyl). When combined with the lower production of CH_3_Cl at pH 10, it is suggested that the ionic form of the syringyl moiety has less photoreactivity than its molecular form when producing CH_3_Cl. This result is consistent with the finding of Yang et al. who took guaiacol as the model compound to study CH_3_Cl formation [18]. However, Dallin et al.’s result was quite different; they found that CH_3_Cl production was inhibited in acidic conditions using syringic acid as the model compound, which demonstrated that the carboxyl form of syringic acid is less reactive than the carboxylate form in producing CH_3_Cl, as the carboxyl form was hard to protonate on the benzene ring [17]. HA is an extremely complex mixture, and there may be many groups that can produce CH_3_Cl besides the lignin-like moieties; thus, the situation with HA is more complicated and cannot be simply analogized by the single model compound. We proposed that the protonation on the benzene ring by hydrogen ion is the crucial procedure for CH_3_Cl production. At pH 4.0, HA was positively charged, and the electrostatic repulsion of H^+^ and the positively charged HA together inhibit the protonation of HA to a certain extent, consequently reducing the generation of CH_3_Cl. However, as the pH increased to 10.0, the functional group in HA was in a less photoreactive ionic form, and thus CH_3_Cl production was lowered again.

### 3.5. Influence of Ferric Ions

In surface waters, ferric ions can complex with organic ligands such as DOM, forming Fe(III)–ligand complexes which can undergo ligand-to-metal charge transfer (LMCT) processes and subsequent Fenton reactions upon irradiation [32], leading to the generation of Fe(II) and reactive oxygen species (ROS) such as HO_2_/O_2_^−^, H_2_O_2_ and ·OH [33]. Therefore, Fe(III) often plays an important role in the fate of environmental pollutants. These experiments were carried out with different concentrations of Fe(III) ranging from 20 μmol L^−1^ to 600 μmol L^−1^ in the presence of 5 mg L^−1^ HA. Considering that ferric species are liable to hydrolyze and form precipitates with increasing pH, in order to get a relatively high reactivity of Fe(III) in the solution, the pH of all the samples containing Fe(III) were adjusted to 4.0 ± 0.1. As shown in Figure 8, the amount of CH_3_Cl produced in the presence of Fe(III) was extremely low. The largest amount of CH_3_Cl in the sample without Fe(III) reached 26.6 ng L^−1^ after 20 h of irradiation, whereas it decreased to less than 7.8 ng L^−1^ with the addition of Fe(III). Therefore, Fe(III) had a significant inhibitory effect on the formation of CH_3_Cl.

The FTIR spectrum of HA with Fe(III) is shown in Figure 6. The major changes of the bands of HA + Fe is as follows: the band of C-O stretching vibration around 1100–1200 cm^−1^ disappeared; the bands of asymmetric and symmetric carboxyl group red-shifted to 1598/1614 and 1404 cm^−1^, respectively; the band near 1600 cm^−1^ indicated that some of the carboxylic acid groups were deprotonated through ligand exchange with Fe(III); and the band at 1400 cm^−1^ could be assigned to –COO–Fe, which was compelling evidence for the HA–Fe(III) complexes [34]. Besides the carboxyl group, phenolic hydroxyl in HA can also complex with Fe(III). It is proposed that the inhibited effect of Fe(III) may be related to the complexing of Fe(III) with a carboxyl group or hydroxyl group in HA [35]. On the one hand, this complexing process reduces the effective concentration of HA in solution by adsorption and co-precipitation (Appendix A); on the other hand, it consumes the functional groups of HA, and to some extent inhibits the protonation of the aromatic ring and disturbs the reaction between the active sites and chloride ions [36]. It should be noted that the results obtained here under acid conditions may not be widely applied in neutral and alkaline conditions. Complexing of HA–Fe(III) significantly affects the photochemical production of CH_3_Cl and the behavior of HA, which needs further investigation.

It is notable that upon analysis of the volatile gas using GC-MS for the sample containing Fe(III) and HA, a significant amount of trichloromethane was confirmed in the scan mode (Appendix A). According to the studies of Liu et al., aromatic halogenated disinfection by-products (DBPs) might be produced from natural organic matter in the presence of chloride during UV irradiation where reactive chlorine species (RCS) acted as the major halogenated reagent, and the resulting DBPs may decompose to aliphatic halogenated DBPs, such as trichloromethane [37]. It can be deduced that there were abundant reactive chlorine species (RCS) in the solution, including radical RCS, e.g., Cl^•^/Cl_2_^•−^, and non-radical RCS, e.g., HClO and Cl_2_. The generation of the radical and non-radical RCS was closely related to ·OH, which could be produced by Fe(III) or Fe(III)–HA (Equation (4)) [38]. The ·OH is capable of oxidizing Cl^−^ to form Cl^•^/Cl_2_^•−^ (Equations (5–7)), Cl_2_ and HClO (Equations (8–9)) [39].
(4)$$\mathrm{Fe}{\left(\mathrm{OH}\right)}^{2+}\;\to {\;\mathrm{Fe}}^{2+}+\;{}^{\cdot }\mathrm{OH}$$
(5)$${}^{\cdot }\mathrm{OH}+{\;\mathrm{Cl}}^{-}\;\to {\;\mathrm{HClO}}^{\cdot -}$$
(6)$${\mathrm{HClO}}^{\cdot -}+H^{+}\;\to {\;\mathrm{Cl}}^{\cdot }+H_{2}O$$
(7)$${\mathrm{Cl}}^{\cdot }+{\;\mathrm{Cl}}^{-}\;\to {\;\mathrm{Cl}}_{2}^{\cdot -}$$
(8)$${\mathrm{Cl}}_{2}^{\cdot -}+{\;\mathrm{Cl}}_{2}^{\cdot -}\;\to {\;\mathrm{Cl}}_{2}+2{\mathrm{Cl}}^{-}$$
(9)$${\mathrm{Cl}}_{2}^{\cdot -}+{}^{\cdot }\mathrm{OH}\;\to \mathrm{HClO}+{\;\mathrm{Cl}}^{-}$$

Although RCS existed abundantly in the solution in the presence of Fe(III), CH_3_Cl was significantly decreased by Fe(III). This result could exclude the effect of RCS on CH_3_Cl formation. The same result has been observed by Yang et al.: RCS was not the active intermediate to generate CH_3_Cl from guaiacol under UV254 irradiation [18].

### 3.6. Generation of Methyl Chloride in Natural Seawater

To investigate the formation of CH_3_Cl in natural seawater, experiments were performed on coastal seawater collected in the Northern Yellow Sea near the Blackrock Reef, Dalian, China. The seawater was first passed through a 0.45 μm membrane and the DOM from the seawater was retained in water. Then, the seawater sample was exposed to light. The results are shown in Figure 9. The concentration of CH_3_Cl after 40 h of irradiation reached 18.8 ng L^−1^ which was at least three times that of the dark control sample. The concentration of CH_3_Cl in the dark control was around 5 ng L^−1^, which can be attributed to the biotic production by some bacteria and microorganisms in the seawater [40,41]. Of course, this biological process may also produce CH_3_Cl under light irradiation through some photosynthetic process. Unfortunately, it is hard to distinguish the biological process from the abiotic photochemical process in this study. Nevertheless, the light process is a significant pathway for the production of CH_3_Cl in natural seawater. Other contributors to the marine source of CH_3_Cl include nucleophilic displacement reactions between CH_3_X and chloride ions, where CH_3_X (X=Br, I) was generated by biological processes of phytoplankton and macroalgae or photochemical process [8,22].

## 4. Conclusions

An investigation of CH_3_Cl production from HA in aqueous chloride solutions and natural seawater under simulated sunlight irradiation revealed the significant photochemical generation of CH_3_Cl in saline waters. The promoting factors for CH_3_Cl generation are light intensity, HA and chloride concentration, whereas the inhibiting factor is ferric ions (under acidic conditions). CH_3_Cl formation was pH-dependent, with the highest amount generated near neutral conditions. Reactions of the kind described in this paper are expected to contribute to CH_3_Cl production in estuarine and coastal waters.

## Figures and Tables

**Figure 1 ijerph-17-00503-f001:**
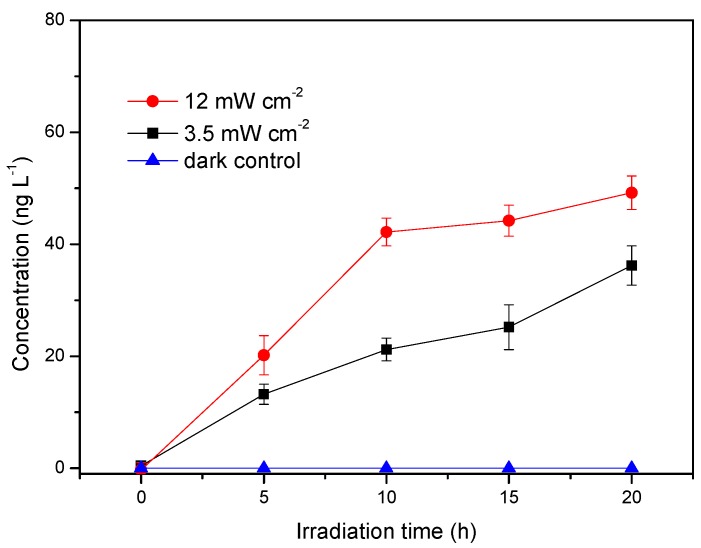
Effect of light intensity on the production of methyl chloride. Reaction conditions: [Cl^−^] = 0.5 mol L^−1^, [HA] = 5 mg L^−1^, pH = 6.5.

**Figure 2 ijerph-17-00503-f002:**
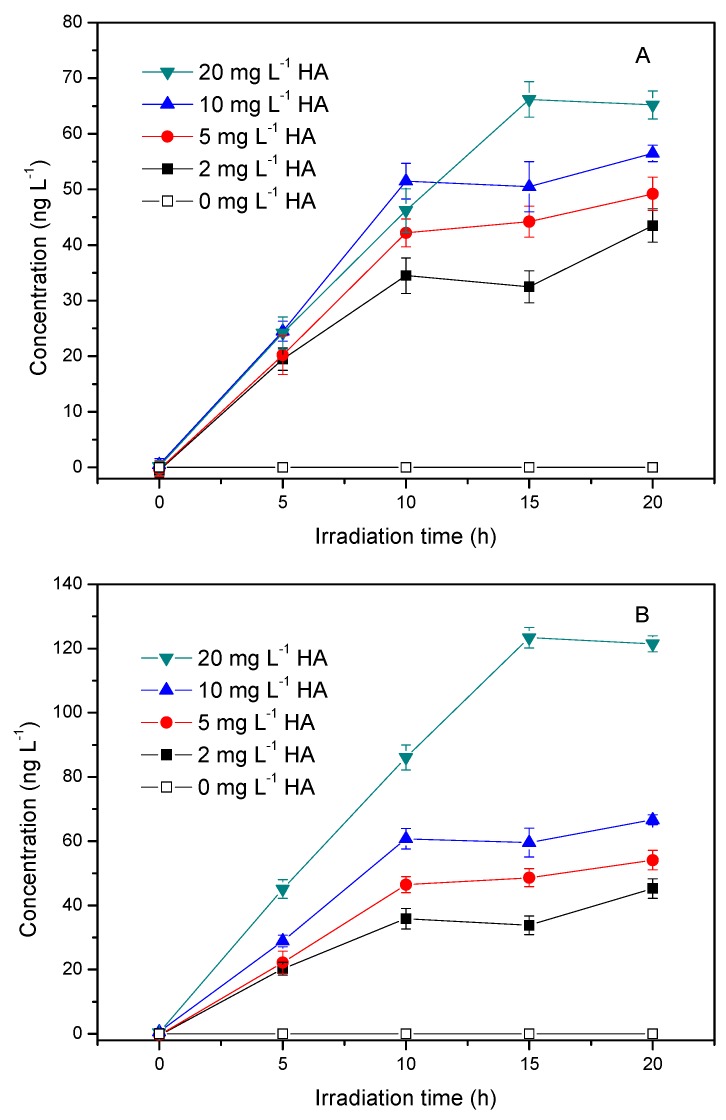
(**A**) Effect of HA concentration on the production of methyl chloride and (**B**) after corrected for light shielding effect using Equations (1)–(3). Reaction conditions: light intensity = 12 mW cm^−2^. [Cl^−^] = 0.5 mol L^−1^, pH = 6.5.

**Figure 3 ijerph-17-00503-f003:**
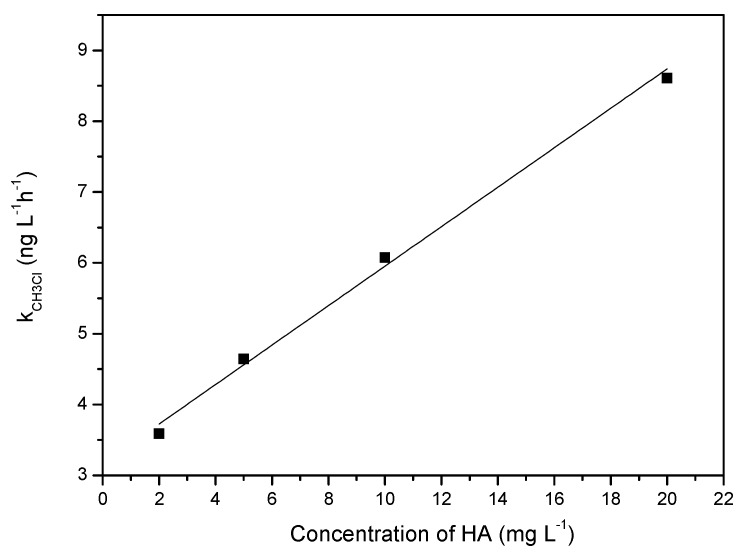
Effect of concentration of HA on the rate constant (k) of CH_3_Cl formation after correction for the light shielding effect. Straight line is the trend line of linear fitting of k_CH3Cl_ and HA concentration. Reaction conditions: [Cl^−^] = 0.5 mol L^−1^, t = 10 h, pH = 6.5.

**Figure 4 ijerph-17-00503-f004:**
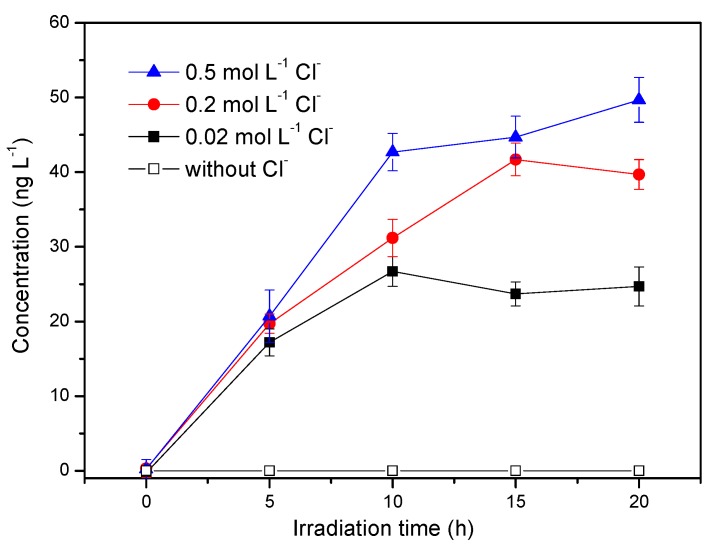
Effect of chloride ion concentration on the production of methyl chloride. Reaction conditions: Light intensity = 12 mW cm^−2^, [HA] = 5 mg L^−1^, pH = 6.5.

**Figure 5 ijerph-17-00503-f005:**
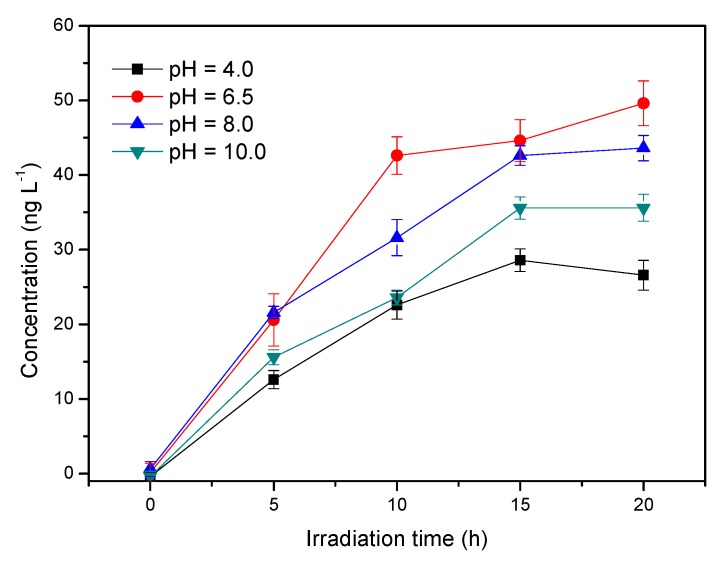
Effect of pH on the production of methyl chloride. Reaction conditions: light intensity = 12 mW cm^−2^. [Cl^−^] =0.5 mol L^−1^, [HA]= 5 mg L^−1^.

**Figure 6 ijerph-17-00503-f006:**
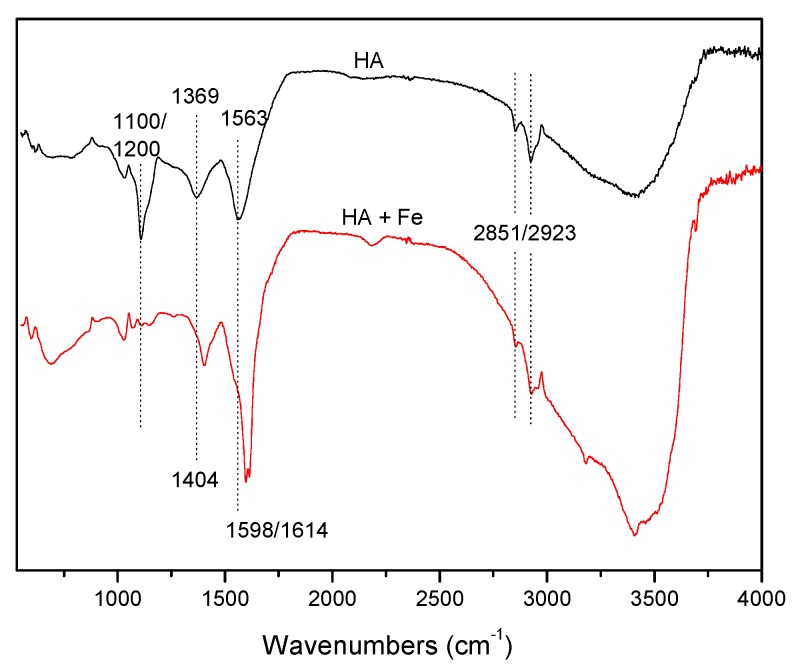
FTIR spectra of HA and HA + Fe. The black curve represents spectrum of HA, and the red curve represents the spectrum of HA + Fe with the corresponding concentrations of 5 mg L^−1^ HA and 100 μmol L^−1^ Fe(III).

**Figure 7 ijerph-17-00503-f007:**
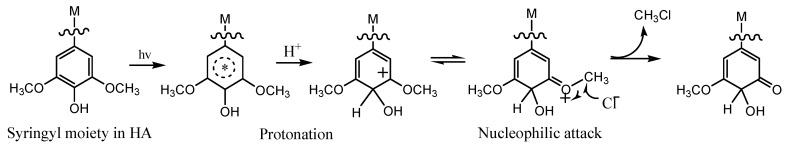
A possible formation pathway of CH_3_Cl from the syringyl moiety of HA under light irradiation.

**Figure 8 ijerph-17-00503-f008:**
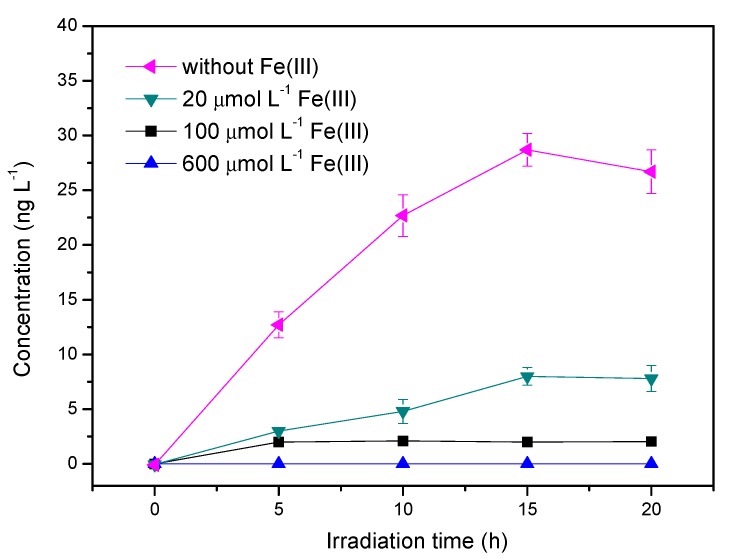
Effect of Fe(III) on the production of methyl chloride. Reaction conditions: light intensity = 12 mW cm^−2^. [Cl^−^] = 0.5 mol L^−1^, [HA] = 5 mg L^−1^, pH = 4.0.

**Figure 9 ijerph-17-00503-f009:**
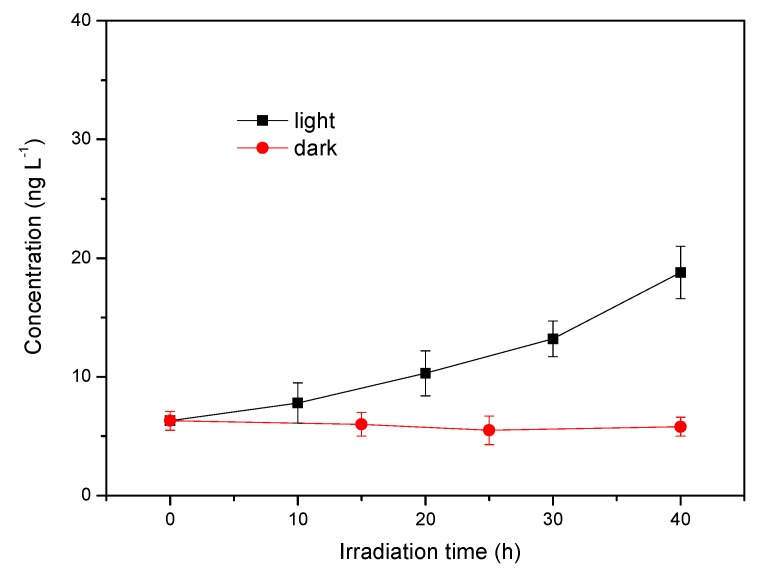
Methyl chloride production in natural seawater from Northern Yellow Sea, China. Reaction conditions: light intensity = 12 mW cm^−2^.

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
