# Peer review of "Photochemical Generation of Methyl Chloride from Humic Aicd: Impacts of Precursor Concentration, Solution pH, Solution Salinity and Ferric Ion"

_ijerph, 2020, doi:10.3390/ijerph17020503_

Round 1

Reviewer 1 Report

In this study, the authors describe a photochemical mechanism that leads to the production of methyl chloride. In the presence of light, only two reactants are required: humic acid and chloride. This is a very interesting finding and may merit publication. However, before recommending the paper for publication, it is my opinion that some additional experiments should be completed, as well as the mechanistic interpretation improved.

As noted above, the key finding of the study is that the illumination of chloride and humic acid under light produces methyl chloride.  I find it very surprising that varying both reactants up to an order of magnitude resulted in no change in the initial rate of the methyl chloride production (e.g., 0 to 5 hr). At a minimum, I think it would be helpful to include a control each in Figure 2 and Figure 3 with no humic acid and no chloride, respectively.

The authors should also address why the kinetics of the reaction slow in each case after about 10 hours, particularly as neither reagent is limiting the initial rate even at much lower concentrations.

In addition, there are a few instances where the authors speculate on explanations for ambiguous or contradictory results that can be resolved with simple measurements or calculations. In my opinion, these should be performed and included in the manuscript prior to publication. Specifically, I recommend the authors address the following:

Line 143: The authors can measure the absorbance of humic acid in these samples and calculate the light shielding effect using standard methods (e.g., those presented in textbooks like Environmental Organic Chemistry by Schwarzenbach et al. or Kinetics of Environmental Photochemistry by Leifer among many other references).

Line 242: Loss of humic acid from the sample can be readily measured by organic carbon analysis and/or UV spectroscopy.

Finally, the authors present some mechanistic interpretation of their data that I think should be improved for clarity and to communicate the key findings.  In the introduction, mechanistic discussion focuses largely on halogen radicals (lines 49-53). While studies that supported other mechanisms are cited, only the halogen radical pathway is described. Consequently, it is surprising and somewhat confusing when, in the results, another mechanism involving a nucleophilic attack of chloride is proposed to better explain the results (line 177-184). Finally, in lines 200-234, halogen radicals are again discussed but later dismissed due to results in the presence of iron. To improve this manuscript, both of these mechanisms should be clearly described in the introduction. The authors should conclude if they have support for one mechanism over the other and provide the evidence for this support.

Reviewer 2 Report

Review Report for "Photochemical generation of methyl chloride from humic aicd: impacts of precursor concentration, solution pH, solution salinity and ferric ion" by Liu et al., 2019. The topic of understanding natural sources of methyl chloride in the environment is important. The manuscript describes a systematic investigation of environmental factors that influence the photochemical production of methyl chloride from humic acid and seawater. The authors independently probe the effect of pH, light intensity, salinity and iron (III) concentration. There are many problems in the experimental design of this article. Thus, the data presented appears to be of no reasonable quality with reaction conditions that mimic natural conditions no sufficiently. The writing quality has a few minor issues, but is reasonably clear. Hence, a few claims need qualifying language and/or additional explanation (see comments below). I recommend accepting the manuscript after it has been thoroughly major revised. Here are my suggestions:

Only HA and NaCl were added to the solution prepared in the experiment, which was not consistent with the general formula of artificial seawater. The effects of other ions in seawater were not considered. Therefore, the results of this experiment could not be discussed by analogy with those obtained by seawater experiment. The author did not state clearly why the formation of CH3Cl in the presence of 20 mg L-1 HA was not obviously higher than lower HA concentrations within 10 h irradiation. If it is speculated that the reason may be related to the shading effect of hyaluronic acid, the mechanism of this effect and its specific way of influence should be elaborated. The pH of the total precipitation of Fe(III) is 3.7. The Fe(III) in the solution prepared by the author with a pH of 4.0 have been completely precipitated and have no reactivity, which cannot affect the generation of CH3Cl in the solution. Therefore, the results obtained by the author are not convincing. In addition, the pH value of seawater is about 8.0. What is the significance of setting such a pH range in this experiment The 0.45 micrometer porous membrane cannot remove all bacteria and microorganisms, and may have the effect of residual biological processes. Therefore it is hard to distinguish the biological production from the abiotic photochemical process. Line 150,to better illustrate the light shield effect, provide the HA absorption and the light emission spectra? Line 151,why set such a salinity range? The average salinity of ocean water is 34.7. Figure 4, CH3Cl is lower at pH8.0, the author should consider the electrostatic repulsion between negative charged HA and chloride anion which could impact the reaction of Cl- and HA. Give more explanation here. Line 254, the author should provide some evidence for the complex of Fe(III)-HA.

9.The following minor points are suggested to improve the readability and clarity of the manuscript.10. Line15, increased change to increasing

Line 55, are the major halogenation reagents Line 57-58 Researches on the … have been concentrated on Line 185 the first step is that the carboxylate form of HA Line 260-261.The 0.45 micrometer porous membrane can remove macroalgae and microalgae, so ,the explanation is wrong. Line 270-277. The entire section needs to be sharper, and reflect the additional analyses required in the preceding sections. Please don't simply copy/paste earlier text in to the conclusion.

Reviewer 3 Report

Overall, it is a pleasant paper. The topic is stimulating so don't be discouraged on the comments received.

Some of the comments need to be taken into account. Some comments can be for the future experiments and scientific questions.

Round 2

Reviewer 1 Report

I am satisfied with the changes the authors have made to the manuscript. 

Author Response

Thank you very much.

Reviewer 2 Report

The comments and suggestions

The paper improved significantly. I have only some minor comments.

Such as,

Line 19. CH3Cl was inhibited—Revise to: the generation of CH3Cl was…

Line 67. …sharp increases of terrestrial DOM…?? It is well known that with the increase of salinity, the amount of dissolved organic matter from marine sources also increases. Not the increase in terrestrial DOM mentioned in the article.

Line 68. Methyl chloride is very volatile, and gas standard of methyl chloride is generally used for calibration. Which company provides the liquid standard you used?

Line 115. How to deal with the background blank? What is the detection limit and precision of the method?

Line 216. The author misquoted the 18th literature. The author of the literature 18 is not Li et al, but Yang et al.

Line 317. 0.45 um membrane filtration cannot exclude the contribution of bacterial production.
